# EBV Association with Lymphomas and Carcinomas in the Oral Compartment

**DOI:** 10.3390/v14122700

**Published:** 2022-12-01

**Authors:** B. J. H. Ward, Danielle L. Schaal, Ebubechukwu H. Nkadi, Rona S. Scott

**Affiliations:** Department of Microbiology and Immunology, Center for Applied Immunology and Pathological Processes, Feist-Weiller Cancer Center, Louisiana State University Health-Shreveport, Shreveport, LA 71103, USA

**Keywords:** Epstein–Barr virus (EBV), latency, viral-associated tumor, Burkitt’s lymphoma, diffuse large B cell lymphoma, Hodgkin lymphoma, head and neck squamous cell carcinoma, oral squamous cell carcinoma, oral hairy leukoplakia, salivary gland lymphoepitheliomas

## Abstract

Epstein–Barr virus (EBV) is an oncogenic human herpesvirus infecting approximately 90% of the world’s population. The oral cavity serves a central role in the life cycle, transmission, and pathogenesis of EBV. Transmitted to a new host via saliva, EBV circulates between cellular compartments within oral lymphoid tissues. Epithelial cells primarily support productive viral replication, while B lymphocytes support viral latency and reactivation. EBV infections are typically asymptomatic and benign; however, the latent virus is associated with multiple lymphomas and carcinomas arising in the oral cavity. EBV association with cancer is complex as histologically similar cancers often test negative for the virus. However, the presence of EBV is associated with distinct features in certain cancers. The intrinsic ability of EBV to immortalize B-lymphocytes, via manipulation of survival and growth signaling, further implicates the virus as an oncogenic cofactor. A distinct mutational profile and burden have been observed in EBV-positive compared to EBV-negative tumors, suggesting that viral infection can drive alternative pathways that converge on oncogenesis. Taken together, EBV is also an important prognostic biomarker that can direct alternative therapeutic approaches. Here, we discuss the prevalence of EBV in oral malignancies and the EBV-dependent mechanisms associated with tumorigenesis.

## 1. Introduction

Epstein–Barr virus (EBV) is a ubiquitous human tumor virus that resides as a lifelong infection in the oral epithelium and B lymphocytes that involves latency and productive viral replication (Figure 1). B lymphocytes support the latent phase of the viral lifecycle exemplified by a restricted viral gene expression program that supports viral evasion of immune surveillance and long-term persistence [1]. Latently infected EBV-positive B lymphocytes can be reactivated into the productive lifecycle, referred to as viral reactivation.

Epithelial cells also support the productive phase of the viral lifecycle following a coordinated viral gene expression cascade resulting in amplification of the viral genome and production of progeny virions [2]. Latency in epithelial cells is observed in carcinomas and in basal tonsillar epithelial cells [3,4].

Infection with EBV is typically acquired asymptomatically in childhood. However, delayed infection until adolescence often causes infectious mononucleosis, a self-limiting lymphoproliferative response to primary EBV infection [5]. Although long-term carriage of the virus is generally benign, EBV is a tumor virus associated with several epithelial and lymphoid malignancies that reflect the tissue tropism of the virus. EBV linked malignancies occur more frequently in immunosuppressed individuals but also develop in immunocompetent persons. EBV-associated cancers include nasopharyngeal carcinoma (NPC), gastric carcinoma (GC), oral squamous carcinoma, Burkitt’s lymphoma, Hodgkin’s lymphoma, diffuse large B cell lymphoma, NK/T-cell lymphomas, and post-transplant lymphoproliferative disease. This review will focus on oral lymphomas and carcinomas associated with EBV infection.

EBV is primarily transmitted through saliva. Early studies showed that infectious EBV is secreted in saliva from patients with infectious mononucleosis and in healthy adults [6]. In healthy individuals, EBV can be detected intermittently over a period of 15 months, with 25% of healthy adults secreting EBV at every sampled time point [7]. The level of viral shedding tends to be stable for days to months and is quickly replaced within 2 min after swallowing [8]. Patients recovering from infectious mononucleosis typically shed higher viral loads for at least 6 months after convalescence [9]. HIV-infected individuals shed higher EBV loads in saliva than HIV-negative individuals. Anti-retroviral therapy to improve the immune status of HIV-infected individuals decreases EBV loads suggesting an additional benefit of anti-retroviral therapy in limiting oral EBV transmission [10]. However, HIV infected individuals on anti-retroviral therapy are at a higher risk of developing an EBV-associated cancer compared to the general population [11].

Epithelial cells in the oral cavity are the major producers of EBV found in saliva [8]. EBV infects and replicates in various epithelial tissues in the oral cavity that include the gingiva, tongue, and tonsillar epithelium [12]. EBV can directly infect an epithelial cell or be transcytosed through polarized oral epithelial cells [13]. Entry is mediated by direct fusion at the plasma membrane or internalized by endocytosis [14]. Various binding interactions between viral glycoproteins and cellular receptors allow for entry into epithelial cells include binding of gH/gL to either integrins, ephrinA2 receptor, and non-muscle myosin heavy chain IIA, viral gB with neuropilin 1. BMRF-2, a viral glycoprotein, can bind to αvβ1 integrins on basolateral surface of polarized epithelial cells, and viral gp350/220 can bind to complement receptor 2 (CR2/CD21) in cases when CR2 is expressed on epithelial cells [15]. Following entry and nuclear delivery of the viral DNA, the linear EBV genome circularizes by recombination of its terminal repeats and is maintained as an extrachromosomal DNA that can be amplified through rolling circle replication [16]. The cell type that produces EBV virions alters the tropism of the virus based on HLA class II interactions with gp42 [17]. Epithelial derived virions have higher levels of the gH/gL/gp42 complex than B cell derived virions, which increases the efficiency of B cell infection [17].

The importance of epithelial cells in the lifecycle of EBV has proven difficult to study due to challenges associated with in vitro epithelial infection. Infection of primary epithelial cells in monolayer culture typically is not efficient and viral DNA is rapidly lost within days [18]. In addition, a non-permissive EBV infection is observed with growth arrest of infected primary nasopharyngeal epithelial cells [19]. In contrast, primary epithelial cells grown in organotypic raft culture, which model the differentiated and stratified layers of the epithelium, indicate that differentiated epithelial cells support robust EBV replication and spread [20]. Differentiation-induced transcription factors (KLF4 and PRDM1) have been identified in regulating the expression of EBV immediate early genes, BZLF1 and BRLF1 [21]. BZLF1 and BRLF1 encode the Z and R transactivators, respectively, that initiate the productive EBV replication cycle. Infection of carcinoma cell lines requires selective pressure to retain the viral genome achieved by using EBV recombinants that carry antibiotic resistance markers and typically results in a latent EBV infection. EBV-positive carcinomas also exhibit a latent EBV infection, which is not typically observed in normal oral epithelial and may reflect the poor differentiation state of tumors. Laser capture microdissection of tonsillar basal epithelial cells detected latent EBV encoded RNAs (EBERs) by quantitative reverse transcription PCR in the absence of EBV immediate early transcripts, supporting the possibility of transient latency in tonsillar basal epithelial cells [4].

EBV infection of circulating resting B lymphocytes in the oral cavity initiates a journey for life-long persistence in the memory B cell compartment. It is also important to note that EBV may directly infect germinal center or memory B cells [22]. Entry into B lymphocytes is mediated by attachment of viral glycoprotein gp350/220 to the complement receptor 2 (CR2/CD21) and interaction of the tripartite glycoprotein complex of gH/gL/gp42 with HLA class II [23]. Endocytosis is required for internalization and gB mediates membrane fusion for viral escape from the acidified endosome. Following viral genome circularization, the viral genome is maintained as an episome [24]. When B cells proliferate and divide, the EBV episome is replicated once per cell cycle by DNA replication licensing at the latent origin of DNA replication, OriP [25]. In the first week after infection, a pre-latent phase is observed with expression of EBV nuclear antigens (EBNAs) and lytic genes [26]. The viral genome is epigenetically silenced using the host epigenetic machinery. DNA methylation and repressive histone modifications deposited on the viral genome restrict viral gene expression [27]. As the EBV-infected B lymphocyte navigates through the B cell maturation/differentiation states, various latency programs are established that support B cell growth and survival of the infected B cells, while avoiding immune surveillance [28]. Infected resting B cells exhibit the growth/latency III program characterized by expression of all EBV nuclear antigens (EBNA-1, -2, -3A, -3B, -3C and -LP), the latent membrane proteins (LMP1, -2A, -2B), and EBV non-coding RNAs: the EBV encoded RNAs (EBERs), *BamHI* fragment A rightward long non-coding transcripts (BARTs) and BART microRNAs. EBV-infected germinal center B cells express the default/latency II program restricted to EBNA1, LMP1, LMP2, and non-coding RNAs. Infected memory B cells that are dividing express only EBNA1 and viral non-coding RNAs (latency 1), while non-viral gene products are detected in non-dividing memory B cells (latency 0). The infected memory B cells provide a long-lived cellular reservoir for EBV persistence and life-long infection. Terminal differentiation of the memory B cell to a plasma cell in response to antigenic stimulation reactivates the EBV productive lifecycle [29] Expression of differentiation-induced plasma cell transcription factors (PRDM1 and XBP-1) activate the expression of EBV immediate early genes BZLF1 and BRLF1 [30].

## 2. Mechanisms of EBV Oncogenesis

EBV infects over 90% of adults worldwide, and EBV is associated with 1% of all cancers [31]. EBV immortalizes B lymphocytes into lymphoblastoid cell lines (LCLs) [32]. However, intrinsic and extrinsic tumor suppression protects the host from developing EBV-associated cancers. Immunosurveillance, an extrinsic tumor suppressor activity, effectively eliminates EBV-infected lymphoblasts, such that immunocompromised individuals are more prone to developing EBV-associated cancers [33]. In contrast, primary epithelial cells are not readily immortalized by EBV without prior genetic alterations [19]. Such transformed epithelial cell become latently infected with EBV in vitro and acquire oncogenic phenotypes similar to what is seen in EBV-associated carcinomas [34]. Thus, the cellular context of genetic and epigenetic alterations can influence the oncogenic outcome in the presence of EBV. EBV infection not only can initiate oncogenic processes as demonstrated by B cell immortalization, but also could contribute to tumor progression and tumor evolution. Thus, EBV likely exerts distinct oncogenic activities depending on the tumor context.

EBV latent infection is a shared feature among EBV-associated malignancies (Table 1). The EBV latent genes have been shown to contribute to various oncogenic phenotypes supported by LMP activation of signaling pathways and by EBNA transcriptional activation that combined act to reprogram host gene expression [35]. In addition to promoting growth, EBV infection and latent proteins contribute to resistance to apoptosis, attenuated responsiveness to differentiation, and enhancement of cellular invasion [36]. Table 2 summarizes known functions encoded by EBV latent genes.

High viral loads due to deregulated viral replication can also contribute to the tumorigenic process. Individuals with infectious mononucleosis typically exhibit higher viral loads in blood and saliva for months after diagnosis and are at an increased risk for developing Hodgkin’s lymphoma [37]. Furthermore, elevated EBV antibody titers frequently precede tumor onset by a few years, a phenomenon observed in EBV-associated Burkitt’s lymphoma, Hodgkin’s lymphoma, and nasopharyngeal carcinoma [38,39]. Although not well understood, the elevated EBV antibody titer likely reflects increased viral loads and viral reactivation. In the context of viral opportunism, higher EBV loads likely increase the risk of infecting premalignant or malignant cells with the infection process inducing a rapid tumor progression.

EBV infection has been shown to increase genomic instability by various mechanisms [40]. EBNA-1 can induce the production of reactive oxygen species promoting DNA damage. LMP-1 inhibits DNA damage responses and DNA repair. EBNA-3C interferes will mitotic spindle checkpoint allowing DNA damage to propagate into the next generation of cells [41]. In addition, EBV infection induces the activation-induced family of cytidine deaminases/apolipoprotein B mRNA editing catalytic polypeptide-like (AID/APOBEC), enzymes with mutagenic activities. AID is responsible for somatic hypermutation (SHM) of B cell immunoglobulin genes undergoing class switch recombination in the germinal center [42]. EBV induced AID activation has been implicated in the translocation of MYC into the immunoglobulin heavy chain or light chain loci, resulting in the overexpression of the c-Myc oncogene. APOBEC deaminates viral RNA/DNA to restrict viral replication but can alter the host genome as well [43].

EBV induces epigenetic alterations that regulate the expression of tumor suppressor expression and apoptotic responses. Nasopharyngeal carcinoma and EBV-associated gastric carcinoma display CpG island hypermethylator phenotypes that silence tumor suppressor gene expression. Common tumor suppressors silenced in NPC and EBV-associated gastric cancer include: RASSF1, CDKN2A/p16, CDH1, and PTEN [44]. Importantly, EBV-associated gastric carcinoma is classified as having an extremely high DNA hypermethylation epigenotype [45]. EBNA-3A and EBNA-3C epigenetically repress BIM, p14, p15, p16, and p18 gene transcription by recruiting polycomb repressive complex 2 [46]. LMP-1 and LMP-2 signaling have been shown to activate the DNA methyltransferases 1, 3A, and 3B, that subsequently leads to tumor suppressor gene silencing [47]. Additionally, LCLs, Burkitt’s lymphoma, and EBV infected B cells display similar tumor suppressor gene silencing due to targeted promoter DNA hypermethylation, despite the observation of global hypomethylation [48]. Alterations in histone modifications are also observed. The repressive histone H3K27 trimethylation mark is elevated in NPC, but not in other EBV associated cancers; while H3K27me3 and H4K20 trimethylation is reduced in EBV-positive lymphoblastoid cells compared to activated EBV-negative B cells [49]. Unlike mutations, epigenetic modification are reversible and identify a potential target for treatment of EBV-associated cancer.

In this review, we will describe various molecular mechanisms utilized by EBV to promote oncogenesis. Specifically, we will direct our attention to EBV-associated lymphomas and carcinomas affecting the tissues and organs of the oral compartment either directly or indirectly. While many of the precise mechanisms remain unclear our goal is to present a current summation of established links between EBV and cancer development and to identify the questions that remain. EBV driven malignancy is a complex process that differs between the various EBV-associated cancer types. It is important to understand both the similarities and differences in the role of the virus across different cancers to prevent and treat EBV-related neoplasms.

## 3. EBV-Associated Oral Lymphoma

Lymphoma is a disease of the lymphatic system characterized by malignant outgrowth of lymphoid cells or lymphoid precursors [50]. Oral lymphomas can arise in the oral cavity or in Waldeyer’s ring, a ring of lymphoid tissue surrounding the oropharynx and nasopharynx. Oral lymphomas are not common, constituting 3% of total lymphomas and 4% of those in patients with AIDS [51]. Although EBV is a potent growth transforming agent of B cells, lymphomagenesis likely results from the complex interaction of viral gene expression and host genetic alterations. The three major types of B cell lymphomas etiologically linked with EBV infection are Burkitt’s lymphoma (BL), Hodgkin’s lymphoma (HL) and diffuse large B cell lymphomas (DLBCL). DLBCL and BL are most commonly observed in the oral compartment, but also frequently at other sites [51]. Manifestations of EBV-associated non-Hodgkin lymphomas such as DLBCL and BL include swelling or ulcerations of the gingiva, tonsils, buccal mucosa, tongue, palate, tooth mobility (alveolar bone loss), and pain. These cancers are often misdiagnosed as symptoms can mimic periapical abscesses/endodontic inflammation.

### 3.1. Burkitt’s Lymphoma

Burkitt Lymphoma (BL) is an aggressive non-Hodgkin B cell lymphoma currently classified into three variants based on both clinical features and cancer epidemiology: Endemic, sporadic and immunodeficiency-associated [52]. Dennis Burkitt initially described endemic BL in equatorial Africa and Papua New Guinea which led to the discovery of EBV as first human tumor virus [53]. Geographically, endemic BL is restricted to regions where malarial transmission is year-round [54]. Endemic BL is a pediatric cancer accounting for 30–50% of all childhood cancers in these regions [55,56,57]. The peak incidence is between 6–9 years of age presenting twice more often in males than females [52]. Endemic BL presents as an extranodal tumor at various anatomical sites that includes the jaw, abdomen, thyroid, kidney, adrenal glands, breast, and ovaries and is almost always positive for the Epstein–Barr virus [55]. Sporadic BL is a rare cancer that occurs throughout the world, appearing 3 times more often in males than females across a wide age group. Sporadic BL comprises 50% of pediatric lymphomas and less than 2% of adult lymphomas [52,58]. EBV is less frequently detected in sporadic BL, with 10–30% of cases being EBV-positive [59]. Although, in some areas such as NE Brazil, EBV association can be as high as 80% [60]. Sporadic BL is three times higher in males than females and frequently manifests within the abdominal region both in lymph nodes and extranodally [61]. Other affected sites include the oropharynx, sinus tract, kidneys, and breast. The third variant of BL was described in persons infected with the human immunodeficiency virus (HIV), and termed immunodeficiency-associated BL. These tumors can present at various anatomical sites as nodal or extranodal tumors [52]. Despite anti-retroviral therapy and maintenance of normal CD4 T cells, immunodeficiency-associated BL constitutes 20–40% of lymphomas in HIV-infected individuals with 30 to 40% of these being EBV-positive [62].

Common to all BL variants is translocation of MYC into the immunoglobulin heavy chain or light chain loci. MYC is placed under the control of the immunoglobulin enhancer resulting in overexpression of the MYC oncogene. Translocation of MYC on chromosome 8 into the immunoglobulin heavy chain locus on chromosome 14 is the most frequent, occurring in 75% of cases [52]. The translocation event results from double strand breaks induced by activation-induced cytidine deaminase (AID). AID deaminates cytosines to uracil resulting in error prone DNA repair and formation of double strand breaks. AID activity is involved in antibody class switch recombination or somatic hypermutation, a process that occurs in germinal center B cells to promote diversity of the antibody pool [63]. Interestingly, EBV infection of resting B lymphocytes has been shown to induce AID, and EBV LMP1 and EBNA3C independently can induce AID expression [42,64]. C-Myc is a transcription factor that as an oncogene promotes proliferation but also apoptosis. Alterations in MYC alone are not sufficient for lymphomagenesis [65]. Additional mutation such as TP53 found in 40% of BL and EBV latency proteins have been suggested to counter the apoptotic activities of c-Myc [66]. In addition, high levels of c-Myc block activation of EBV lytic replication preventing chromatin looping interactions between the lytic origin of replication and the BZLF1 promoter [67].

EBV is detected in the majority of endemic BL cases carrying a clonal EBV genome in tumor cells indicating that viral infection preceded cellular transformation [68]. How EBV contributes to the BL lymphomagenesis is still being defined due to challenges posed by studying a human pathogen in animal models. However, loss of the viral genome renders many BL cell lines sensitive to apoptotic cell death [69]. EBNA1 and the EBV non-coding RNA expressed in BL have been shown to mediate cell survival. EBNA1 interaction with the deubiquitinase USP7, known to stabilize p53 and Mdm2, lowered p53 levels and protected cells from p53-mediated apoptosis [70]. The transforming capacity of EBNA1 in promoting lymphoma have yielded conflicting results in transgenic mice due to inherent differences in the mouse strain and expression construct used [71]. Instead, pharmacological inhibition of EBNA1 as a druggable target has shown growth inhibition in vitro and anti-tumor activities in animal xenografts [72]. Reintroduction of EBER1 noncoding RNA into BL lines that lost EBV restored apoptotic resistance by increasing BCL2 activity [73]. Similarly, re-introduction of the EBV BART derived miRNAs protected BL cells that lost EBV from apoptosis by directly interfering with CASP3 [74].

The malarial *Plasmodium* parasites have been epidemiologically linked to endemic BL. The mechanistic interplay between malaria, EBV, and BL is complex and still unresolved. Repeated malarial infections have been suggested to impair cytotoxic T cell responses potentially affecting immune control of EBV [75]. Chronic malaria in children is associated with increased EBV viral loads that can also promote tolerized/exhausted T cell responses to EBV-infected B cells [76]. Loss of immune surveillance is also consistent with the higher EBV-positivity in immune-associated BL. Another role of malaria in promoting BL may involve B cell activation and aberrant activation of AID [77]. *Plasmodium falciparum* infected mice exhibited abnormal expression of AID in B cells apart from the reactions of the germinal center [78]. Extracts from *P. falciparum* infected red blood cells were also shown to enhance AID transcription and protein levels in tonsillar B cells [79]. Importantly, consecutive infections of p53 deficient mice with *Plasmodium chabaudi* induced mature B cell lymphomas with AID-dependent translocations that included MYC/IgH [80]. Thus, EBV and malarial infections are likely synergistic interactions that lead to the rapid progression of BL in children. From these observations, the presence of EBV in BL is suggested to confer resistance to apoptosis and enhance genomic instability through AID activation.

Whether EBV is essential for the development of BL has been questioned by the prevalence of EBV-negative sporadic BL and immune-associated BL cases. However, two recent studies identified EBER-negative cases that where EBV microRNAs were detected, raising the possibility that EBV-negative BL may be derived from EBV-infected B cells [81]. An alternate possibility is that EBV is present early in lymphomagenesis but mutations that complement EBV oncogenic activity allow for viral loss [82]. Such genetic changes can be seen in EBV-negative BL having a higher frequency of mutations that interfere with apoptosis (TP53 and USP7) and mutations in TCF3 and ID3 enhance B cell receptor signaling [52]. However, differences in mutational landscape between EBV-positive and -negative BL argues that these tumors may be distinct entities. EBV-positive BL show increased AID expression, an AID mutational signature, higher mutation load, but few driver mutations [83]. Mutations in chromatin modifiers such as Swi/Snf complex subunits ARID1A and SMARCA 4 were more frequent in EBV-positive BL [84]. Together these observations support EBV selecting for distinct genetic and epigenetic alterations that contribute to lymphomagenesis. EBV also serves as an important biomarker to guide prognostic outcome and future therapeutic approaches.

### 3.2. Diffuse Large B Cell Lymphoma

Diffuse large B cell lymphoma (DLBCL) is the most common non-Hodgkin’s lymphoma, accounting for 30% of all lymphoma cases [85]. DLBCL comprise a collection of heterogeneous and aggressive cancers that present at extranodal sites at the time of diagnosis. In the oral compartment, DLBCL manifests in Waldeyer’s ring and salivary glands. Other anatomic sites involved include gastrointestinal tract, bone, spleen, testes, thyroid, liver, and the kidneys [86]. Molecular classification based on mutation and gene expression profiles has grouped DLBCL into various subtypes based on features of the cell of origin. Two main subtypes are the germinal center B cell (GCB) DLBCL and activated B-cell like (ABC) DLBCL [85]. GCB DLBCL has features of germinal center B cells, while ABC DLBCL has features of post-germinal center, plasmablast B cells. In the DLBCL classification scheme, EBV-positive DLBCL not otherwise specified (NOS), previous referred to as EBV- positive DLBCL of the elderly, has been classified as a separate clinical subtype [87]. The “not otherwise specified (NOS)” is used to exclude the diagnosis of other EBV-associated large B cell tumors such as primary effusion lymphoma, EBV-positive plasmablastic lymphoma or EBV-positive mucocutaneous ulcer.

EBV-positive DLBCL (NOS) are rare malignancies that typically present in immunocompetent individuals over 50 years of age, but this cancer can also affect children and younger adults. EBV-positive DLBCL (NOS) accounts from 3–15% of DLBCL cases, being more prevalent in Asia and Latin America [88]. EBV-positive DLBCL (NOS) has two subtypes: monomorphic and polymorphic subtypes. The monomorphic subtype is more frequently associated with advanced age and carries a poor prognosis [86]. EBV-positive DLBCL (NOS) tumors are derived from a clonal B cell expansion as noted by a clonal immunoglobulin rearrangement in up to 60% of cases [89]. EBV positivity is based on EBER detection by in situ hybridization methods in greater than 80% of the malignant cells. However, various studies have used different thresholds for EBER positivity with cut-off values as low as 10% to define EBV-positive DLBCL (NOS). The significance of partial EBV association is not well understood and could suggest a “hit-and-run” mechanism where genetic changes and viral epigenetic reprogramming compensate for loss of the viral genome. Alternatively, malignant EBV-positive cells may support oncogenic phenotypes of EBV-negative cells through cell communication or by establishing an immunosuppressive tumor environment. In the infected tumor cells, EBV exhibits a type II/III latency expression pattern, with latency III being more frequently detected [90]. Other EBV positive lymphomas that arise due to immunodeficiency (e.g., post-transplant lymphoproliferative disease) also display a type III latency pattern, suggesting that diminished T cell surveillance may fail to control the EBV-infected B cells. Indeed, PDL1 and PDL2 ligands are increased in EBV-positive DLBCL (NOS) as a mechanism that can inhibit T cell anti-tumor responses [91].

EBV-positive DLBCL (NOS) display unique molecular features compared to EBV-negative DLBCL. Overall, EBV-positive DLBCL (NOS) have a lower mutational burden suggesting that EBV can replace mutations otherwise required to drive the oncogenic phenotype [92]. Consistent with this, EBV-positive DLBCL (NOS) display fewer MYC, BLC2, and BL6 rearrangements [93]. The recurrent mutations detected in EBV-positive DLBCL (NOS) affected NF-kB, IL6/JAK/STAT, and WNT signaling pathways, known to be activated by expression of EBV latency proteins [94]. Loss of function mutations in MYD88, CD79B, and CKDN2A are also frequently observed in EBV-negative DLBCL but not present in EBV-positive DLBCL (NOS). Moreover, EBV-positive DLBCL have a higher frequency of mutations involving chromatin regulators (ARID1A, KMT2A, KMT2D, TET2, and DNMT3A), indicating genome-wide alterations in epigenetic regulation of gene expression [92,95]. Deletions at chromosome 6q, which encompasses regions that encode PRDM1 required for plasma cell differentiation and the anti-inflammatory TNF induced protein 3 (A20) also occurred in EBV-positive DLBCL (NOS). PRDM1 is a differentiation-dependent transcription factor that is required for transcriptional activation of the EBV immediate early genes, BZLF1 and BRLF1 [96]. The loss of PRDM1 not only disrupts B cell terminal differentiation into plasma cells but may also help maintain EBV in a latent state. Altogether, the distinct mutation landscape in EBV-positive DLBCL (NOS) enforces the notion that EBV guides the etiology of this unique DLCBL entity. Understanding the molecular features in various age groups and geographical regions will identify novel therapeutic approaches to improve the poor prognosis associated with EBV-positive DLBDL (NOS).

### 3.3. Hodgkin Lymphoma

Hodgkin lymphoma is a B cell malignancy characterized by the presence of few malignant cells surrounded by many non-neoplastic inflammatory cells. Based on histological and immunophenotype presentation, Hodgkin’s lymphoma (HL) is classified into two major types: known as classical HL and nodular lymphocyte predominant HL. In classical HL, the malignant cell is the Hodgkin and Reed/Sternberg cell (HRS). In nodular lymphocyte predominant HL, the malignant cell is the lymphocyte predominant cell (LP) [97]. As nodular lymphocyte predominant HL is rarely associated with EBV, this review will discuss only the classic HL type.

Classical HL account for 90% of all HL. The HRS malignant cell is derived from a germinal center B cell that has undergone somatic hypermutation of immunoglobulin genes and clonal expansion [98]. The HRS cell is described as large (>50 micron diameter), multinucleated cells containing an eosinophilic cytoplasm and infected by EBV [99]. HL tumors typically affect the lymph nodes but can also occur at extranodal sites. Oral manifestations occur at the tonsils, tongue, Waldeyer’s ring, and palate [100] HL accounts for approximately 10% of all lymphoma cases diagnosed. HL is more common in males and the most prevalent cancer among 15- to 19-year-old adolescents [101].

EBV is associated with approximately 20 to 50% of classical HL cases. The presence of the virus in HRS cells is typically identified via in situ hybridization (ISH) to EBV DNA or EBV EBER RNA. Among the classical HL subgroups, EBV is common in the nodular sclerosis and mix cellularity classical HL subtypes, with rare detection in lymphocyte rich or depleted types [102]. An EBV latency II pattern is observed in HRS tumor cells. An increased risk for EBV-positive HL has been observed following infectious mononucleosis [103]. HL incidence is significantly elevated in HIV-infected individuals despite anti-retroviral therapy, and most cases are EBV-positive [104]. These observations suggest a causal link between EBV and escape from immune surveillance in the development of classical HL.

Although the HRS cells originate from germinal center B cells carrying clonal immunoglobulin rearrangements [98], these HRS tumor cells have lost several B cell markers and B cell receptor signaling. Crippling mutations are detected in the immunoglobulin genes that rendered the B cell receptor (BCR) nonfunctional [105]. All cases with crippling BCR mutations are EBV-positive, suggesting that EBV may rescue HRS cells with nonfunctional BCR mutations [106]. Loss of BCR activity also can result from transcriptional silencing of BCR expression, downregulation of B cell specific transcription factors (OCT2, BOB1, and PU1), and upregulation of repressors of B cell genes [107]. A functional BCR is required for B cell survival and to escape elimination following somatic hypermutation in the germinal center. Thus, a pivotal event in classical HL oncogenesis is the acquisition of mechanisms that counter the loss of BCR signaling. As such, HRS tumor cells have various mutations that promote constitutive activation of JAK/STAT signaling and NF-kB signaling.

EBV latency factors has been suggested to compensate for the loss of BCR and support tumor growth and survival. High levels of LMP1 are detected in classical HL. LMP1 mimics CD40 signaling and activates JAK/STAT, PI3K, and NF-kB signaling pathways [108]. Forced expression of LMP1 in germinal center B cells transcriptionally reprogrammed gene expression similar to that of HRS tumor cells [109]. LMP1 was also shown to disrupt B cell terminal differentiation into plasma cells by downregulating PRDMI/BLIMP1 [110]. LMP2 mimics BCR signaling and can directly substitute for the loss of BCR in HRS tumor cell [111,112]. However, LMP2 requires many of the downstream BCR signaling factors that are absent in HRS tumor cells. Thus, LMP2 may act independently of BCR signaling by activating PI3K signaling and reprogram B cells to support cell growth and survival. In addition, EBV latent gene products can also alter immune cell activities and promote tumor microenvironments conducive to oncogenesis. LMP1 suppresses cytotoxic T cell responses via the induction of PD-L1 expression [113]. EBV BART microRNAs as well as other EBV gene products can be packaged and released by cellular exosomes to affect distant cells [114]. Exosomes derived from EBV-positive classical HL were reported to alter TNF-α and IL-10 cytokine profiles of host macrophages [115].

The tumor microenvironment in classical HL is composed of inflammatory cell infiltrates suggested to create an immunosuppressive environment for HRS cells to escape immune surveillance and/or support the growth and survival of HRS cells [116]. T cells are a major component. Activated CD4-positive T helpers cells (Th1 and Th2) and regulatory T (Tregs) cells are present without detection of CD8-positive cytotoxic T cells or NK cells [117]. Activated Th1 cells displaying CD40L and CD30L markers were suggested to support growth and survival of HRS cells, while Th2 and Tregs cells maintained an immunosuppressive environment. Compared to EBV-negative classical HL, EBV-positive cases display increased number of regulatory T cells, and tumor associated macrophages likely mediating a more immunosuppressive tumor environment that prevents immune-mediated elimination of the tumor cells [118].

Although EBV has been implicated in supporting HRS and development of HL, the presence of EBV-negative HL has suggested multiple avenues that can lead to lymphoma. EBV may substitute for mutations and promote oncogenic phenotypes that support growth, survival, and immune evasion. EBV-positive classical HL display distinct genetic alterations from EBV-negative classical HL, similar to what is observed in EBV-positive DLBCL [119]. Whether EBV-negative classical HL has a distinct etiology from EBV-positive cases is possible. However, detection of traces of EBV in EBER-negative cases or loss of the viral genome following epigenetic reprogramming suggest a role for EBV in the early stages of HL lymphomagenesis [120].

## 4. EBV-Associated Oral Carcinomas

Over 90% of oral cancers are squamous cell carcinomas [121]. Oral squamous cell carcinoma (OSCC) is a type of head and neck squamous cell carcinoma (HNSCC) that arises from the epithelial mucosa lining the oral cavity and oropharynx. In the oral compartment, EBV is associated with lymphoepitheliomas of the salivary glands and is the causative agent in the benign hyperplastic lesion known as oral hairy leukoplakia. Although EBV is often detected in carcinomas from the oral cavity and oropharyngeal tract, incomplete association of EBV in histologically similar tumors has questioned the viral contribution in OSCC. The following section will describe EBV in oral carcinomas, oropharyngeal carcinomas, lymphoepitheliomas of the salivary gland and oral hairy leukoplakia.

### 4.1. Oral Squamous Cell Carcinoma

Oral squamous cell carcinomas (OSCC) comprise a heterogenous group of carcinomas that arise at various anatomical sites in the oral cavity. Oral cavity squamous cell carcinoma (OCSCC) arises from lips, anterior tongue, roof and floor of mouth, gingiva, and buccal mucosa. Oropharyngeal squamous cell carcinoma (OPSCC) involves the tonsils, posterior third of tongue, and soft palate. OPSCC due to infection with human papillomavirus has rapidly emerged as a distinct entity and will be discussed separately.

OSCC accounts for an estimated 2.5% of all cancers worldwide [122]. Based on various epidemiological studies, various carcinogenic factors have been implicated in the development of OSCC that include chewing and smoking tobacco, alcohol consumption, chewing betel quid, environmental pollution, poor oral hygiene, sun exposure in lip cancer, and infection with human papillomavirus [123]. OSCC occurs more frequently in males than females and individuals over the age of 60. Immune competency is an important factor in OSCC as HIV-infected individuals develop OSCC earlier in life [124]. Diagnosis of OSCC occurs often at advanced stages of disease with evidence of regional or distant metastasis. In the United States, the relative 5-year survival is estimated at 68%, but improved survival up to 86% is observed in patients with localized tumors [125]. However, treatment modalities (surgery, radiation and chemotherapy) often leave patients with severe morbidities affecting their appearance and ability to eat, drink, and speak [126]. Thus, identifying biomarkers for early detection and new therapeutic approaches are needed to reduce the morbidity and mortality of OSCC worldwide.

EBV has been detected at higher levels in OSCC compared to healthy tissues with many studies detecting EBV DNA, RNA and protein in tumor cells [127,128]. However, an etiological role for EBV in OSCC is still debated due to an incomplete viral association. The oncogenic process in OSCC is complex involving various cofactors and alternative genetic pathways that combined increase the risk for cancer. As OSCC is comprised of a heterogenous assortment of cancers, the molecular features and subtypes are still being defined [129]. Studies comparing the genomic landscape in EBV-positive and -negative OSCC are needed to provide a better understanding of the drivers in these cancers.

The viral gene expression program in EBV-positive OSCC is not well established. Various studies have detected EBER expression in OSCC tumor cells [130,131]. Although detection of EBERs is the standard for assigning EBV positivity, it is unclear if EBERs are a reliable marker in OSCC. EBER expression pattern in OSCC can be patchy and weak with a mixture of nuclear and cytoplasmic signals. Epithelial differentiation may influence EBER expression with reduced levels noted in more differentiated areas of NPC tissue [132]. Whether EBERs are a reliable marker in OSCC warrants further evaluation. In support of an association, other EBV markers such as EBNA2 and LMP1 protein levels have been detected in OSCC at higher levels than normal tissues. LMP1 positivity was also detected at higher levels in dysplastic tissue than in OSCC, implicating EBV in the early stages of tumor progression [130,133].

In the oral cavity, the microbiome can influence the prevalence of herpesviruses (EBV, HCMV, HSV) as well as the degree of inflammation [134]. In chronic periodontitis, a significant increase in EBV-infected gingival epithelium was observed compared to healthy gingival tissue [12]. Mechanistically, presence of bacteria such as *Porphyromonas gingivalis* and *Porphyromona endodalis* have been shown to produce butyric acid, an HDAC inhibitor that can reactivate latent EBV [135]. Such microbial interactions may increase EBV viral loads and promote a pro-inflammatory environment. High EBV viral loads provide EBV the opportunity to infect hyperplastic/dysplastic cells and contribute to the progression to carcinoma.

Although EBV infection is not sufficient to immortalize epithelial cells in culture, viral infection can confer oncogenic features to already immortalized/transformed cells lines similar to what is observed in EBV-associated NPC and gastric carcinoma. EBV infection of hTERT-immortalized gingival epithelial cells induced a DNA hypermethylator phenotype [136], which is also observed in NPC and EBV-associated gastric carcinoma tumors. Both LMP1 and LMP2A can induce DNA methyltransferases (DNMTs) and subsequent silencing of tumor suppressor genes [130]. Expression of LMP2A is sufficient to interfere with epithelial differentiation in spontaneously transformed HaCaT cells [137]. Moreover, EBV-infected epithelial cells displayed an attenuated response to differentiation [136,138]. In addition, EBV-infected epithelial cells acquire an invasive cellular phenotype that correlates with the metastatic nature of NPC [34,139]. LMP1 expression is sufficient to enhance cellular motility and migration via activation of PI3K, NF-kB and ERK-MAPK signaling pathways in NPC [140,141,142,143]. These EBV-induced oncogenic phenotypes are retained following loss of the viral genome suggesting a role for EBV-induced viral epigenetic modifications in these processes [34,136,139].

### 4.2. HPV-Positive Oropharyngeal Squamous Cell Carcinoma

Reductions in smoking and alcohol consumption has decreased OSCC incidence worldwide. Nonetheless, the past thirty years has seen the emergence of OPSCC associated with human papillomavirus (HPV) infection in developed countries [144,145,146]. HPV is the leading cause of cervical carcinoma worldwide and a causal agent for various anogenital carcinomas. However, the cases of HPV-positive OPSCC have equaled and will surpass cases of cervical carcinoma in developed nations [147,148,149]. Ninety percent of HPV-positive OPSCC are due to HPV16 [150,151]. HPV infection is rarely detected in OCSCC [152]. A shift in the demographics of HPV-positive OPSCC have also been observed. Historically HPV-positive OPSCC appeared more frequently in white males in their late 50s and 60s [153]. However, in the past decade, HPV-positive OPSCC includes older age groups (> 70 years of age) with marked increases observed in females and racial groups [146]. HIV-infected individuals show an increased risk for HPV-positive OPSCC with higher frequency of recurrence and worse outcomes despite anti-retroviral therapy.

The majority of HPV-positive OPSCC occur at the tonsils or base of tongue, lymphoid rich regions where EBV also resides. Moreover, HPV-positive OPSCC has distinct clinical features not evident in other HPV associated cancers [151]. HPV-positive cervical cancer results in a slower progression to carcinoma with early HPV productive lesions detected. In contrast, HPV-positive OPSCC lack these typical premalignant lesions and appear to progress rapidly such that patients are often diagnosed at advanced stages. Despite the advanced staging of the tumor, HPV-positive OPSCC have better outcomes than HPV-negative cases. These features have classified HPV-positive OPSCC as a distinct entity and suggest a role for additional cofactors in tumor development.

As a potential cofactor, EBV is detected in 5–20% of HPV-positive OPSCC [154,155,156,157]. Specific capture of tumor cells devoid of lymphocytic infiltrates detected EBV EBER transcripts by reverse-transcription PCR in 25% of tonsillar and 80% of HPV-positive base of tongue tumors [158]. Using EBER in situ hybridization in the same patient cohorts proved less sensitive but identified 20% of base of tongue tumors as being co-infected [158]. As a correlate to HPV-positive OPSCC, EBV has also been detected in HPV16 and HPV18-positive cervical carcinomas (ranging between 12–80% EBV positive) being more often detected in cervical carcinoma than in low grade intraepithelial lesions [159,160,161]. EBV was also more frequently found in cervical carcinomas with integrated HPV [160,162]. Interaction of EBV and HPV in the epithelium may have unintended effects on each viral lifecycle that combined may accelerate tumor development.

Although the contribution of EBV to HPV-positive OPSCC carcinogenesis still needs to be defined, similar mechanisms suggested for EBV driven NPC carcinogenesis may be involved. EBV is a well-established etiological agent in nasopharyngeal carcinoma (NPC), a HNSCC occurring in the nasopharynx. The proximity to the oropharynx and presence of a lymphoid infiltrate in NPC tumors provide EBV access to nasopharyngeal epithelial cells [163]. Early genetic changes can predispose dysplastic cells to latent EBV infection rather than the productive lytic replication that typically occurs in the epithelium. Loss of CDKN2A and overexpression of cyclin D1 have been shown to facilitate stable, latent EBV infection of nasopharyngeal epithelial cells [19]. EBV-positive carcinomas generally exhibit a latency II viral gene expression program (EBNA1, LMP1, LMP2, BARF1, EBERs, BART miRNAs and long noncoding RNAs). Evidence of clonal expansion is also observed based on EBV episomes having unique number of terminal repeats as a maker of clonality, where the length of the terminal repeats inversely correlates to LMP1 and LMP2 expression [164,165,166].

HPV-immortalization of epithelial cells may create an environment that favors latent EBV infection rather than productive replication. Epithelial interactions between EBV and HPV have been studied in epithelial organotypic raft culture, a physiologically relevant culture producing stratified and differentiated epidermal layers that support HPV and EBV productive lifecycle. HPV-immortalized keratinocytes grown in organotypic rafts inhibited EBV replication following de novo infection [167]. An abortive pattern of infection was observed where the immediate early transactivator BZLF1 was expressed in the absence of replicative factors. A slight increase in EBER levels was also noted [167]. The HPV16 oncogene E7 was sufficient to block EBV replication [167]. One of the functions of E7 is to facilitate the proteasomal degradation of the retinoblastoma pocket proteins (pRb, p107, and p130) [168]. Recent studies show that EBV replication within differentiated epithelium requires pRb (Myers et al., submitted). pRb is a tumor suppressor that inhibits cell cycle progression at the S-phase checkpoint, and deregulation of these pathways may enhance the oncogenic potential of EBV. A separate study indicated a different outcome when EBV infection preceded HPV immortalization. In this case, HPV appeared to increase the maintenance of EBV latent genomes and enhanced EBV lytic reactivation in organotypic rafts [169]. The different outcomes between the studies suggest that the outcome of EBV infection may depend on the order of infection, the physical and epigenetic state of the viral genome, and/or EBV-induced epigenetic alterations that support viral replication. Such EBV-induced epigenetic reprogramming of epithelial cells was shown to attenuate epithelial differentiation and enhance cellular invasiveness [136,138,139].

At the molecular level, HPV-positive OPSCC display distinct genetic changes compared to HPV-negative OSCC [170]. HPV16-positive OPSCC typically have high levels of p16, which is often used as a surrogate marker for HPV-positivity in OPSCC [171,172,173]. p16 is an inhibitor of cell cycle progression that prevents cyclin-dependent kinases 4 and 6 (CDK) phosphorylation of the retinoblastoma pocket protein, pRb. HPV16 E7 directly inactivates retinoblastoma pocket protein family bypassing the regulatory p16 checkpoint [174]. In addition, HPV-positive OPSCC typically have wildtype TP53 that is functionally inactivated by HPV E6. When TP53 mutations occur in HPV-positive OSCC, these tumors carry a poorer prognosis. TP53 is frequently mutated in HPV-negative OPSCC [175]. HPV-positive OPSCC also exhibit an apolipoprotein B mRNA editing enzyme catalytic polypeptide (APOBEC) mutational signature. The APOBEC cytosine deaminase is an innate antiviral response to viral infection that is elevated in HPV-infected cells [176]. Other less frequent mutations in HPV-positive OSCC include *PIKCA*, *ZNF750*, *CASZ1*, *PTEN < CYLD*, and *DDX3X*, while mutations in *FAT1*, *CDKN2A*, *NOTCH1*, *CASP8*, and *HRAS* are associated with HPV-negative OSCC [175]. The distinct genomic landscape of HPV-positive and HPV-negative OSCC likely influences the outcome of EBV infection supporting the virus as a cofactor in the progression of these cancers.

### 4.3. Oral Hairy Leukoplakia

Oral hairy leukoplakia (OHL) are benign oral lesions that presents as elevated white patches on the dorsum and lateral borders of the tongue, and in rare cases the soft palate, pharynx, or esophagus are involved [177]. Other features include hyperkeratosis, epithelial hyperplasia, ballooning degeneration, acanthosis, and mild or moderate inflammatory infiltrate [178]. OHL affects individuals with severe immunodeficiency such as acquired immunodeficiency syndrome (AIDS) due to HIV infection, organ transplant recipients, and following chemotherapy. Treatment for OHL is important as these lesions can be precursors to squamous cell carcinomas [179]. OHL can occur in up to 50% of individuals presenting with AIDS and occurs more frequently in males than females [180]. Some rare cases of OHL have been reported in HIV-negative individuals on long term steroid treatments [181]. Diagnosis is based on histopathological findings and the presence of EBV using in situ hybridization (EBER-ISH) [182]. However, EBER RNAs are not typically expressed in OHL, with EBER-ISH detecting the high viral DNA content in the lesions [183].

OHL is a result of productive EBV replication within oral epithelial cells. EBV infects oral squamous epithelial cells and replicates in the absence of cell-mediated immunity. Manifestation of OHL is one of the earliest signs of AIDS, and OHL is strongly correlated with decreased CD4+ counts [184]. A deficiency of Langerhans cells has been reported in OHL tissues that may allow escape from immune surveillance [185]. Langerhans cells are antigen-presenting immune cells required for mounting an immune response against viral infection in the epithelium. Furthermore, circulating EBV-positive monocytes have been detected in HIV-positive individuals [186]. These infected monocytes have been shown to enter the epithelium and allow for infection of keratinocytes [186]. Antiviral treatment with nucleoside analogs (acyclovir, valacylovir, and famicyclovir) resolves these lesions in several weeks [187]. HIV-positive individuals receiving treatment to improve immune function and antiretroviral treatment reduces OHL as well, but recurrence is common

Although OHL is associated with lytic EBV replication, several latently associated factors (except the EBERs) are expressed simultaneously with lytic genes [188]. EBNA2 is expressed in almost all OHL cases [189]. LMP1 and the viral Bcl2 homolog, BHRF1, are also expressed to promote cell survival and stimulation of epithelial proliferation. LMP1 induces anti-apoptotic genes such as A20 and activates NF-kB, MAPK, and JNK signaling, while BHRF1 plays a role in delaying epithelial differentiation as well as inhibiting apoptosis [190]. Altogether these transforming genes extend the life of infected cells and promote the unique characteristics of OHL. A distinct attribute of OHL is co-infection with multiple different EBV types and strain variants [191]. EBV type 1 and type 2 are identified by sequence variations in EBNA2. Sequence differences at LMP1 has identified several variant strains. Based on LMP1 genotyping, OHL lesions can be infected with multiple variant strains [192]. Superinfection of multiple EBV strains has been suggested to be due to transcription of EBNA2 which upregulates expression of EBV receptor, CR2/CD21 [193].

Although EBV growth promotion is frequently associated with latent gene products, OHL epithelial hyperplasia implicates replicative gene products in promoting carcinogenesis. This is supported by B cell immortalization being less efficient in the absence of BZFL1. Presence of BZLF1 also enhanced lymphoma development in humanized mice models [194]. Several paracrine factors, such as VEGF growth factor, IL6, and IL10 cytokines, are produced in cells undergoing lytic replication to modulate the tumor microenvironment to support the growth of the tumor cells [195].

### 4.4. Lymphoepitheliomas of the Salivary/Parotid Gland

Lymphoepitheliomas are undifferentiated carcinomas associated with a dense lymphoid infiltrate that occur at various anatomical sites being associated with EBV. These epitheliomas resemble undifferentiated NPC also having a lymphoid infiltrate. Salivary gland lymphoepitheliomas involving the parotid gland are rare oral tumors that are associated with EBV. A high incidence is observed among native people in Greenland, Alaska, and some Asian populations [196]. In these endemic areas, all tumors are associated with EBV, while tumors arising in nonendemic areas are sporadically associated with EBV [197]. Histological and pathological characteristics are comparable to undifferentiated nasopharyngeal carcinoma and association of EBV is described as a latent infection [198].

EBV EBERs are detected strictly in malignant epithelial cells and not significantly detected in the infiltrating lymphocytes or surrounding benign tissues [196]. Detection of EBV in salivary tumors is determined by EBER expression in tumor cells and detection of the viral DNA by PCR [199]. LMP1 protein has been detected in a subset of salivary gland epitheliomas and LMP1 sequence variants are often found in the endemic population [200].

## 5. Conclusions

In the more than 55 years since its discovery, EBV has been shown to play a complex role in the pathogenesis of several human malignancies. Though EBV associated oral cancers are rare, the oral cavity is central to the life cycle of the virus within its human host. The oral anatomy provides proximity between the B lymphocytes and epithelial cells in which the virus completes its life cycle. The persistence of EBV within the tissues of the oral cavity is a risk factor for cancers in the oral compartment and at other anatomical sites. The current understanding of EBV mediated oncogenesis is summarized in Figure 2. EBV can directly alter apoptosis, proliferation, and other major signaling pathways (NF-kB, JAK-STAT). However, a commonality observed among the EBV-associated cancers is the existence of additional oncogenic co-factors. Examples include host genetic alterations or the presence of additional pathogens (HPV, HIV, Malaria). EBV-mediated perturbation of the host may complement the effect of these co-factors, or EBV may provide an alternate route to the evolution of the tumor cell. EBV is not always present in histologically similar tumors, but genetic and epigenetic alterations are different when comparing EBV-positive and negative states. As potent epigenetic modifier, EBV manipulation of the host epigenome provides another oncogenic mechanism through reprogramming of host gene expression. Such epigenetic changes can be retained following loss of the viral genome as a viral mechanism for “hit-and-run” oncogenesis. In addition, EBV has a demonstrated ability to alter the tumor microenvironment and interfere with host immune surveillance. Future studies will continue to elucidate and define the role of EBV in tumorigenesis and establish EBV as prognostic factor in cancer. Better control or elimination of EBV infection with vaccination will likely reduce the burden of EBV-associated cancers.

## Figures and Tables

**Figure 1 viruses-14-02700-f001:**
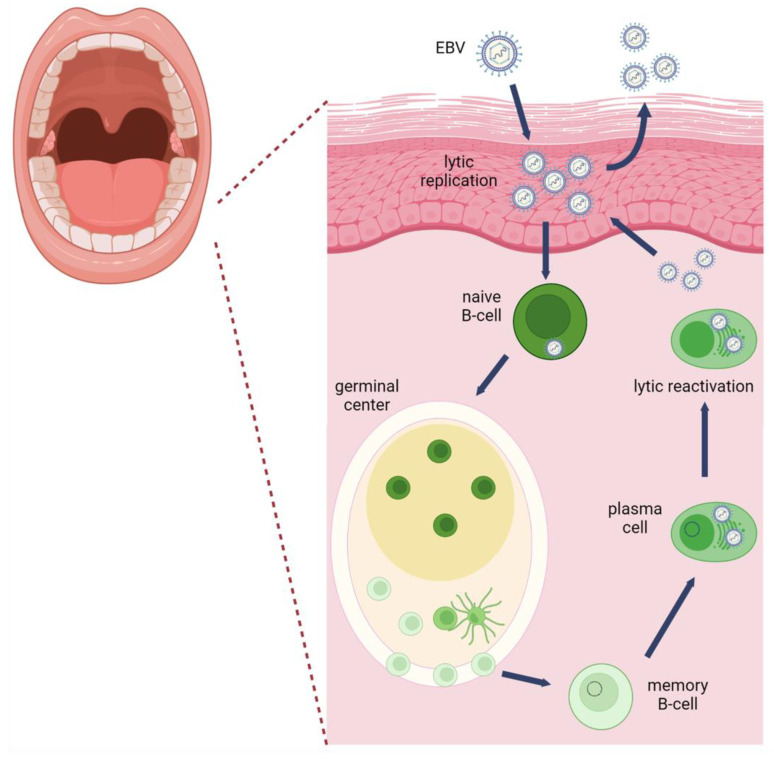
Life cycle of EBV within the oral cavity. EBV is transmitted via saliva and infects oral epithelial cells. Epithelial cells support productive viral replication in the upper differentiated layers of the epithelium. In B cells, EBV adopts various latency programs that support B cell maturation/differentiation and promote B cell growth and survival. As long-lived cells, memory B cells provide a life-long reservoir for EBV as a latent infection. Differentiation of memory B-cells to plasma cells stimulates EBV reactivation to produce new progeny virions that infect the epithelium for shed in saliva or re-infection of other naïve B-cells. Image was created with BioRender.

**Figure 2 viruses-14-02700-f002:**
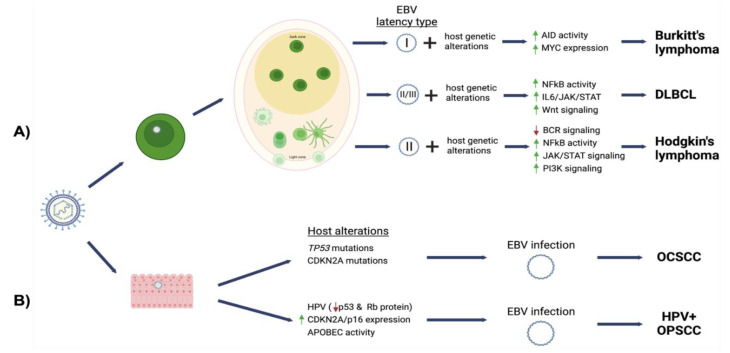
Models summarizing the contribution of EBV in oral cancers. (**A**) EBV Infected B cells circulate to secondary lymphoid tissues where viral latent gene products promote clonal expansion and germinal center survival and differentiation into centroblasts. Tumors may arise from cells participating in GC (BL) reactions or post-GC selection (cHL, DLBCL NOS). The viral latency program complements acquired host mutations and influences the outgrowth of the tumor cell. In BL, EBV infection can induce AID which can increase genomic instability. AID overexpression can induce MYC chromosomal translocations into the immunoglobulin enhancer region leading to MYC overexpression. EBNA 1 has been suggested to interfere with p53 to prevent MYC induced apoptosis. In DLBCL, EBV-positive tumors tend to have a lower frequency of mutations that EBV-negative types. Recurrent mutations activate NF-kB/JAK-STAT/Wnt signaling pathways. In HL, EBV may support growth and survival following loss of B cell receptor signaling (BCR), with constitutive activation of NF-kB, and JAK/STAT signaling pathways conferring resistance to apoptosis. EBV LMP1 induces pathways activated in HL and EBV + DLBCL. (**B**) The oral epithelium supports the EBV productive lifecycle. Genetic changes that interrupt EBV productive replication and/or facilitate EBV latent infection have been linked to development of OSCC. In oral cavity squamous cell carcinomas (OCSCC), mutations in TP53 and CDKN2A are drivers of OCSCC tumor progression. EBV infection likely occurs at a later stage where additional genetic and epigenetic alterations allow for long term latent infection. EBV infection and expression of latent gene products can influence cell growth, survival, migration, and differentiation. The majority of oropharyngeal squamous cell are due to HPV infection. Expression of the HPV oncogenes E6 and E7 facilitates the degradation of p53 and the retinoblastoma family of pocket proteins, respectively. Presence of HPV induces APOBEC and genomic instability that contributes to the evolution of the tumor cells. HPV-immortalization due to E7 expression interferes with EBV productive replication that result in abortive or latent infections. Increased persistence of EBV and expression of EBV gene products in HPV-immortalized cells may contribute to the rapid progression observed for HPV-positive OPSCC. Image was created with BioRender.

**Table 1 viruses-14-02700-t001:** EBV prevalence and viral latency program in associated oral cancers and lesions.

Cancer	Cellular Origin	% EBV Associated	EBV Expression Pattern	Latent Gene Products Detected
Burkitt lymphoma Endemic Sporadic HIV-related	Germinal center centroblast	~100% 10–80% 30–40%	Latency I	EBERs, EBNA1, BARTs
Hodgkin lymphoma-Classic variant Nodular sclerosis Mixed cellularity HIV-related Hodgkin lymphoma-NLPHL variant	Post-germinal center centroblast	40–50% 10–40% 70–80% >90% rare	Latency II	EBERs, EBNA1, LMP1, LMP2, BARTs
DLBCL NOS HIV-related, centroblastic HIV-related, immunoblastic	Post-germinal center centroblast	3–50% * 30% 90%	Latency II/III Latency I/II/III Latency I/II/III	Depends on latency program
Oral Squamous Cell Carcinoma	Epithelial Cells	0–80%		EBERs, EBNA2, LMP1
HPV-positive oropharyngeal squamous cell carcinoma	Epithelial cells	5–25%		EBERs
Oral Hairy Leukoplakia	Epithelial Cells	100%	Lytic	All viral genes
Salivary Gland Epithelioma	Epithelial Cells			EBERs, LMP1

* prevalence varies geographically, NLHPL—Nodular lymphocyte-predominant Hodgkin lymphoma, DLBCL—Diffuse large B cell lymphoma, NOS—not otherwise specified, HIV—human immunodeficiency virus, EBNA—EBV nuclear antigen, LMP—latent membrane protein, EBER—Epstein–Barr virus (EBV)—encoded small RNAs, BARTs—BamHI A rightward transcripts.

**Table 2 viruses-14-02700-t002:** Overview of functions encoded by EBV latent genes.

EBV Latent Gene Product	Function
EBNA1	Required for viral genome latent replication and segregation Promotes resistance to apoptosis by degradation of p53 Increases ROS production and genomic instability
EBNA2	Essential for B cell immortalization Regulates viral and host gene expression by interacting with host transcription factors and EBNALP Regulates chromatin looping and accessibility
EBNA3A/C	Recruits polycomb repressor complex 2 for epigenetic repression of cyclin dependent kinase inhibitors (CKIs) and apoptotic factors Induces AID expression (EBNA 3C) Promotes bypasses cell cycle checkpoints that increase proliferation and genomic instability
EBNA3B	Tumor suppressor activity
EBNALP	Transcriptional co-activator of EBNA2
LMP1	Mimics CD40 receptor signaling Activates NF-kB/MAPK/JAK-STAT/PI3K signaling Induces DNA methyltransferase activity Promotes proliferation and survival Induces AID expression Immune modulation
LMP2A/B	Mimics host B cell receptor (BCR) signaling Blocks tyrosine kinase signaling following antigen activation of BCR Inhibits viral reactivation Induces DNA methyltransferase activity Enhances cell migration Inhibits epithelial differentiation
EBER1	Abundantly expressed viral RNA in EBV latency Confers resistance to apoptosis Retains cellular ribosomal factor L22 in the nucleoplasm Blocks interferon inducible protein kinase R (PKR)-mediated inhibition of protein synthesis
EBER2	Binds and recruits Pax5 to EBV terminal repeats
BART microRNAs	Increases resistance to apoptosis

## Data Availability

Not applicable.

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
