# Peer review of "EBV Association with Lymphomas and Carcinomas in the Oral Compartment"

_viruses, 2022, doi:10.3390/v14122700_

Round 1
Reviewer 1 Report
In the manuscript entitled “EBV and Oral Cancer” the authors discussed about the different aspects of EBV infection that can potentially lead to the development of cancer. The manuscript is nicely drafted and easy to understand for the readers. The authors did a good job in citing previous literatures and in overall its good and informative. There are few minor concerns that need to be addressed.
1. Although the manuscript is very informative but I feel the authors stressed on EBV infection in different cells including B cell. So the title seems to be misfit with the content. The authors described about the different types of B cell lymphomas and that does not have to do anything with oral cancer. I recommend to rephrase the title and make it more generalized or otherwise the authors should specifically shorten the sections that deals with B cell lymphomas.
2. In the very 1st sentence in the introduction authors mentioned about the lifelong infections in oral cavity and oropharynx. But in general, in B cells also they can be latent for life long. So it will be good to incorporate all the other type of cells that can also have latent infections.
3. In Line 34-37 the authors mentioned “B lymphocytes support the latent phase of the viral lifecycle exemplified by a restricted viral gene expression program that supports viral evasion of immune surveil-35 lance and long-term persistence”. Anyone who is not directly related to EBV field will assume that B cell only support latent infection. But B cells can be also induced to lytic reactivation. it’s better to rephrase the statement.
3. In Line 74 the authors mentioned about viral BMRF1. The authors should introduce these viral genes before going into any further details.
4. Under the subheading “Mechanisms of EBV oncogenesis” the authors discussed some of the important pathways by which EBV influence cellular processes and induce oncogenesis. The authors should also discuss about the myc translocation. They did that in the later section but this is worth mentioning here.
5. There is one typos in Line 327. DLBCL is misspelled as DLCBL.
With these corrections I highly recommend this manuscript for publication.
Author Response
We first would like to thank the reviewer for the helpful comments to clarify and improve this review. We have addressed the points as detailed below. The layout of the tables was also modified, and we added some missing references that were notice upon proofreading. We have tracked all the changes to the manuscript.
- Although the manuscript is very informative but I feel the authors stressed on EBV infection in different cells including B cell. So the title seems to be misfit with the content. The authors described about the different types of B cell lymphomas and that does not have to do anything with oral cancer. I recommend to rephrase the title and make it more generalized or otherwise the authors should specifically shorten the sections that deals with B cell lymphomas.
As we were asked to write a comprehensive review, we included EBV-positive B cell lymphomas that manifest in the oral cavity. We feel that shortening the lymphoma section would remove important information. Instead, we have amended the title to “A Review on the Association of Epstein-Barr Virus With Oral Cancers” as recommended by reviewer 2.
- In the very 1stsentence in the introduction authors mentioned about the lifelong infections in oral cavity and oropharynx. But in general, in B cells also they can be latent for life long. So it will be good to incorporate all the other type of cells that can also have latent infections.
We have rewritten the first paragraph to clarify this point as well as point 3. We have included the statement that EBV is latent in carcinomas and in the basal tonsillar epithelial cells.
- In Line 34-37 the authors mentioned “B lymphocytes support the latent phase of the viral lifecycle exemplified by a restricted viral gene expression program that supports viral evasion of immune surveil-35 lance and long-term persistence”. Anyone who is not directly related to EBV field will assume that B cell only support latent infection. But B cells can be also induced to lytic reactivation. it’s better to rephrase the statement.
We have changed the first paragraph to include that latently infected B cells can be induced to reactivate the virus into the productive lifecycle.
- In Line 74 the authors mentioned about viral BMRF1. The authors should introduce these viral genes before going into any further details.
Line 74 (now line 85) referred to BMRF2 which is a viral glycoprotein. We have included this information as suggested.
- Under the subheading “Mechanisms of EBV oncogenesis” the authors discussed some of the important pathways by which EBV influence cellular processes and induce oncogenesis. The authors should also discuss about the myc translocation. They did that in the later section but this is worth mentioning here.
We have included the following sentence: “EBV induced AID activation has been implicated in the translocation of MYC into the immunoglobulin heavy chain or light chain loci, resulting in overexpression of the c-Myc oncogene”.
- There is one typos in Line 327. DLBCL is misspelled as DLCBL.
Thank you for catching and we have corrected the typographical error.
Reviewer 2 Report
In this review, the author summarizes the association of Epstein-Barr virus (EBV) with oral cancer. EBV infections are typically asymptomatic and benign; however, the latent virus is associated with multiple lymphomas arising in the oral cavity. This is a well-written review describing association of EBV in oral lymphoma like Burkitt’s lymphoma and Hodgkin’s lymphoma. It is clearly a big, concerted effort and well written review, which sheds a new light on details on host alteration factors during EBV infection. Despite of all positive outcomes of the review, I found several points, where it can be improved.
Comments –
1. The title of the review is EBV and oral cancer. Can Author change the title “Review on Association of Epstein-Bar virus with oral cancer”. Something like that, so that it will be more representable. I know paper is available on the same title, so Author can use review term.
2. Author can show the illustration figure of EBV genome with life cycle, so that mechanism of the EBV oncogenesis will be more clear
3. In 38 and 39, dot is in next line after reference [3]. Please correct it.
4. In line 161, under reference [35], an underline is visible. Please correct it.
5. In line 615, under some, an underline is visible. Please correct it.
6. In line 954 and 1447, Author uses the same reference. “Kataoka, K.; Miyoshi, H.; Sakata, S.; Dobashi, A.; Couronné, L.; Kogure, Y.; Sato, Y.; Nishida, K.; Gion, Y.; Shiraishi, Y.; et 1447 al. Frequent structural variations involving programmed death ligands in Epstein-Barr virus-associated lymphomas. 1448 Leukemia 2019, 33, 1687-1699, doi:10.1038/s41375-019-0380-5.”
Author Response
We first would like to thank the reviewer for the helpful comments to clarify and improve this review. We have addressed the points as detailed below. The layout of the tables was also modified, and we added some missing references that were notice upon proofreading. We have tracked all the changes to the manuscript.
- The title of the review is EBV and oral cancer. Can Author change the title “Review on Association of Epstein-Bar virus with oral cancer”. Something like that, so that it will be more representable. I know paper is available on the same title, so Author can use review term.
Thank you for the suggestion and we have changed the title as suggested- "A Review on the Association of Epstein-Barr Virus with Oral Cancers"
- Author can show the illustration figure of EBV genome with life cycle, so that mechanism of the EBV oncogenesis will be more clear
We are not sure what changes to make in regard to this comment and have not altered the illustrations.
- In 38 and 39, dot is in next line after reference [3]. Please correct it.
Thank you for catching and we have corrected the typographical error.
- In line 161, under reference [35], an underline is visible. Please correct it.
We apologize but we were unable to locate the underline on our copy.
- In line 615, under some, an underline is visible. Please correct it.
We apologize but we were unable to locate the underline on our copy.
- In line 954 and 1447, Author uses the same reference. “Kataoka, K.; Miyoshi, H.; Sakata, S.; Dobashi, A.; Couronné, L.; Kogure, Y.; Sato, Y.; Nishida, K.; Gion, Y.; Shiraishi, Y.; et 1447 al. Frequent structural variations involving programmed death ligands in Epstein-Barr virus-associated lymphomas. 1448 Leukemia 2019, 33, 1687-1699, doi:10.1038/s41375-019-0380-5.”
Thank you for catching this. We had inadvertently duplicated the reference list and have made this correction.